# Hoof slip duration at impact in galloping Thoroughbred ex-racehorses trialling eight shoe-surface combinations

Kate Horan [1]*, James Coburn[2], Kieran Kourdache[3], Peter Day[1], Henry Carnall[2☉], Liam Brinkley[2☉], Dan Harborne[2☉], Lucy Hammond[3], Sean Millard[1], Renate Weller[4], Thilo Pfau[1,4]

**1** The Royal Veterinary College, Hertfordshire, United Kingdom, **2** James Coburn AWCF Farriers Ltd, Newmarket, United Kingdom, **3** The British Racing School, Newmarket, United Kingdom, **4** University of Calgary, Calgary, Alberta, Canada

☉ These authors contributed equally to this work.
* khoran@rvc.ac.uk

**Data Availability Statement:** All relevant raw data are available in figshare at https://doi.org/10.6084/m9.figshare.26865769.v1.

## Abstract

Horseshoes used during racing are a major determinant of safety as they play a critical role in providing traction with the ground surface. Although excessive hoof slip is detrimental and can predispose to instabilities, falls and injuries, some slip is essential to dissipate energy and lower stresses on the limb tissues during initial loading. This study aimed to quantify hoof slip duration in retired Thoroughbred racehorses galloping over turf and artificial (Martin Collins Activ-Track) tracks at the British Racing School in the following four shoeing conditions: 1) aluminium; 2) steel; 3) GluShu (aluminium-rubber composite); and 4) barefoot. High-speed video cameras (Sony DSC-RX100M5) filmed 389 hoof-ground interactions from 13 galloping Thoroughbreds at 1000 frames per second. A marker wand secured to the lateral aspect of the hoof wall aided tracking of horizontal and vertical hoof position in Tracker software over time, so the interval of hoof displacement immediately following impact (hoof slip duration) could be identified. Data were collected from leading and non-leading forelimbs at speeds ranging from 24–56 km h$^{-1}$. Linear mixed models assessed whether surface, shoeing condition or speed influenced hoof slip duration (significance at p≤0.05). Day and horse-jockey pair were included as random factors and speed was included as a covariate. Mean hoof slip duration was similar amongst forelimbs and the non-leading hindlimb (20.4–21.5 ms) but was shortest in the leading hindlimb (18.3±10.2 ms, mean ± 2.S.D.). Slip durations were 2.1–3.5 ms (p≤0.05) longer on the turf than on the artificial track for forelimbs and the non-leading hindlimb, but they were 2.5 ms shorter on the turf than on the artificial track in the leading hindlimb (p = 0.025). In the leading hindlimb, slip durations were also significantly longer for the aluminium shoeing condition compared to barefoot, by 3.7 ms. There was a significant negative correlation between speed and slip duration in the leading forelimb. This study emphasises the importance of evaluating individual limb biomechanics when applying external interventions that impact the asymmetric galloping gait of the horse. Hoof slip durations and the impact of shoe-surface effects on slip

**Funding:** This research was funded by the Horserace Betting Levy Board, project 4497, Prj786, grant titled 'S.A.F.E.R.' (Shoe Assessment for Equine Racing).

**Competing interests:** I have read the journal's policy and the authors of this manuscript have the following competing interests: J.C. owns the company James Coburn AWCF Farriers Ltd., which employed H.C., L.B. and D.H. at the time of the study. J.C., P.D., H.C., L.B. and D.H. and are now registered farriers. T.P. and R.W. are the owners of EquiGait, a provider of gait analysis products and services. This does not alter our adherence to all polices on sharing data and materials. The funders had no role in the design of the study or the collection, analyses and interpretation of data.

were limb specific. Further work is needed to relate specific limb injury occurrence to these hoof slip duration data.

## Introduction

The slip duration of horses' hooves upon contact with the ground surface can affect their performance, orthopaedic health and risk of injury [1, 2]. Both too much and too little hoof slip can cause injury. Most racehorse fractures arise from an imbalance between microdamage and repair due to repeated cyclic loading [3] but the magnitude of the impact forces is also important and linked to lameness and injury incidents [4, 5]. Energy dissipation during hoof slip is important, as it serves to lower the rate that the longitudinal ground reaction force is applied on the limb in question and, in turn, this means lower forces and less stress are placed on limb tissues during initial loading [6–10]. Having some hoof slip at impact also constrains bending moments on the cannon bone [11]. In addition, high frequency oscillations at impact can increase the risk of damage to subchondral bone and joint tissues [12–16], and it is important that there is some hoof slip to mitigate this. Moderate longitudinal hoof sliding can also improve performance, by increasing stride length [1]. However, it is important to recognise that excessive hoof slide can predispose to injury, such as tears to the digital flexor muscles [17]. Therefore, to prevent injuries linked to excessive slip and biomechanical instability prior to loading, slip distances and durations must be constrained by having some traction at the hoof-ground interface.

If ground surface conditions or a horse's shoeing condition do not offer sufficient traction or, alternatively, inhibit slip and decrease the rate of energy dissipation, injury risk could increase in either scenario. Ground surface is a significant risk factor for injuries to racehorses [18–22]. For example, surface type has been implicated as a trigger factor for altering superficial digital flexor tendon loading and joint kinematics [23]. Surface properties also influence hoof vibrations, accelerations and ground reaction forces, with accelerations and forces typically being reduced on synthetic surfaces compared to turf and dirt surfaces [24–26]. Furthermore, although forelimbs are generally more likely to fracture than hindlimbs, fracture patterns amongst limbs can be surface dependent with, for example, hindlimbs more commonly fracturing on turf than dirt tracks [22]. Epidemiological data also suggest that certain shoe-types, such as those used in the United States with high toe grabs, rims or pads, which increase grip, are associated with a higher risk of racehorse injury [27–31]. With these considerations in mind, there is increasing interest in quantifying surface conditions at racetracks, and the use of horseshoes is tightly regulated in most countries, including the United Kingdom. The British Horseracing Authority (BHA) currently enforce that horses running in flat races conducted on turf enter the parade ring fully shod except where the BHA has consented before the Declaration to Run is submitted or in exceptional circumstances when the Stewards give permission. In addition, the following shoe types are prohibited: shoes which have protrusions on the sole other than calkins or studs on the hind, with the latter limited to 3/8 inch in height; American type toe-grab plates; and shoes with a sharp flange [32]. Nonetheless, to date, there has been limited research quantifying the effect that different ground surfaces and shoeing conditions have on hoof slip duration, particularly in galloping horses. Although the high-speed field kinematics of hoof contact have been quantified in horses galloping on an artificial track [33], the magnitude of hoof slip under different shoe-surface conditions has not specifically been assessed. Other studies considering shoe-surface implications for hoof slip have

tended to focus on slower gaits. However, slip duration data from horses trotting on concrete in different shoeing conditions [2], trotting on grass [1] or stone dust tracks [34], cantering on grass with/without studs [35], or from an ex-vivo model trialling different surface or shoe-types [36, 37], may not be readily applicable to live racehorses galloping on grass and artificial surfaces, barefoot and with shoes devoid of protrusions from the sole.

The purpose of the current study was to quantify hoof slip durations in retired racehorses as they galloped over grass and artificial training tracks whilst barefoot or wearing steel, aluminium or rubber-composite shoes. In the UK, most horse races are run on turf but training takes place on both turf and artificial surfaces. Therefore, the surfaces selected for this study reflect typical UK training and racing tracks, and the shoeing conditions reflect both common shoeing practices (aluminium in racing; steel in training) and readily accessible options (barefoot and rubber-composite). We hypothesised that slip durations would be longest on turf and for the barefoot condition, based on 12 months of BHA race data, which showed that there was an increased risk of a horse slipping in flat turf conditions if partially shod [38]. In addition, as hoof accelerations were previously found to show a speed-dependent response to shoe and surface combinations in this sample population [26], we were also interested to investigate how shoe-surface condition might impact slip duration across different gallop speeds.

## Materials and methods

### Ethics

Ethical approval for this study was received from the Royal Veterinary College Clinical Research Ethical Review Board (URN 2018 1841–2), which included a written consent form for horse owners and jockeys.

### Experimental animals

A convenience sample of 13 retired Thoroughbred racehorses at the British Racing School (BRS) in Newmarket, UK, were included in this study. The horses were in regular work, including gallop training, and were normally utilised for jockey education. They ranged in age from 6–20 years old, with heights from 1.6–1.7 m, and they had masses between 421 and 504 kg. The horses were also included in previously published studies [26, 39–41], and further details on individual horse body dimensions and hoof morphometrics are available [39]. All horses were considered sound by the jockeys, farriers and BRS senior management prior to data collection, and they are regularly checked by a veterinarian. Details on jockey experience and training for the four participating jockeys have previously been published [39, 41]. During trials, horse—jockey pairings were fixed, while shoe—surface conditions varied. One horse was ridden by two jockeys, giving rise to 14 possible horse—jockey pairings. Trials took place across multiple days for each horse-jockey pair to acquire data for as many of the eight possible shoe—surface combinations as was feasible; limitations were imposed due to horse and jockey availability and routine turf accessibility restrictions implemented by the BRS to avoid 'hard' going [26, 39–41]. The shoe-surface combinations completed by each horse-jockey pair are summarised in [39], but please note that video footage was not available for one horse and therefore 14 (rather than 15 horse-jockey pairs, as per [39]) are included in the current study.

### Experimental design

**Trial conditions.**   The horse—jockey dyads underwent randomised data collection trials on level artificial and turf surfaces in the following four shoeing conditions: (1) aluminium race-plates (Kerkhaert Aluminium Kings Super Sound horseshoes); (2) barefoot; (3) GluShus

(aluminium—rubber composite horseshoes); and (4) steel shoes (Kerkhaert Steel Kings horse-shoes). Details on the trimming and shoeing protocol maybe found in [41]. Typical shoe masses were 134 ± 26 g (mean ± 2 S.D., unless otherwise stated) for the aluminium shoes (n = 67), 191 ± 50 g for the GluShus (n = 56), and 333 ± 11 g for the steel shoes (n = 65). The artificial surface used was the Martin Collins Activ-Track, which comprises sand and CLOPF fibre. It is wax-coated, dust-free and designed for use in all weather conditions. Turf conditions during data collection ranged from 'soft' to 'good-firm'. Full details of the weather on and preceding data collection days have previously been published [39] and information regarding the adaptation period, warm-up period and exercise trials can also be found in previous publications [26, 41].

**Equipment and filming.** This study took a similar approach to previous work capturing hoof slip with high-speed video [1, 24, 33, 42]. The horses were filmed using four high-speed video cameras (Sony DSC-RX100M5) at 1000 frames per second, for an interval of approximately 3 s. The cameras were spaced 3.5 m apart, at a height of 75 cm; an arrangement that ensured the overall capture of at least one hoof strike per limb in each gallop run. The total field of view was approximately 15 m. We filmed on a straight section of each track, approximately 200 m from the start point. This study required a visual cue from which to track hoof motion in the sagittal plane. Custom-made hoof marker wands were therefore created, similar to previous studies [1, 24], with a design that ensured they projected above the ground level even on soft surfaces. They consisted of two wooden sticks glued together at 90 degrees, supporting white polystyrene balls that could be easily detected when filming at approximately 8.5 m away from the horse and jockey (Fig 1A; [41]). The hoof wands were secured to the lateral aspect of the right fore and right hind hooves of each horse using Superfast hoof adhesive [41], because we were filming from the right hand side. The central marker on the wand was tracked unless it was obscured in the video, in which case the upper left marker was used. Jockeys were additionally provided with a GPS device (Holux RCV 3000) to carry in their pocket during trials. This device recorded their position every second, and from these data, speed during gallop runs could be quantified.

## Data processing

Video data for 389 slip events from 207 gallop runs were available for processing from the 13 horses (14 horse—jockey pairs) testing the eight possible shoe—surface combinations. This incorporated 93 slip events from the leading forelimb, 107 slip events from the non-leading forelimb, 88 slip events from the leading hindlimb, and 101 slip events from the non-leading hindlimb. Occasionally, there were trials that did not generate any viable data due to the hoof marker wand breaking or becoming obscured by dirt kicked up by the horse, or because the horse ran close to the grass verge on the artificial track where the wand was out of view. There were also two trials where slip duration data were discounted because the horse was bucking or had become disunited.

Hoof slip duration reflects the time from when the hoof first contacts the ground surface (Fig 1B) until it enters the weight-bearing period of the stance phase and its position becomes largely fixed. By tracking the vertical trajectory of the hoof wand, the precise point at which the hoof contacted the ground could be identified; time point 1 (Fig 1C and 1D). The vertical trajectory of the marker was also used to help identify the point at which the hoof stabilised; time point 2 (Fig 1B and 1C). Emphasis was placed on evaluating the vertical rather than horizontal trajectory of the marker over time, simply because it had a more consistent trace at the point of entry into the weight-bearing phase. However, sometimes it was still challenging to identify time point 2 if there was not a clear inflexion point at the transition into mid-stance, and sometimes the hoof bounced on the turf surface (Fig 1D). Therefore, care was taken to always

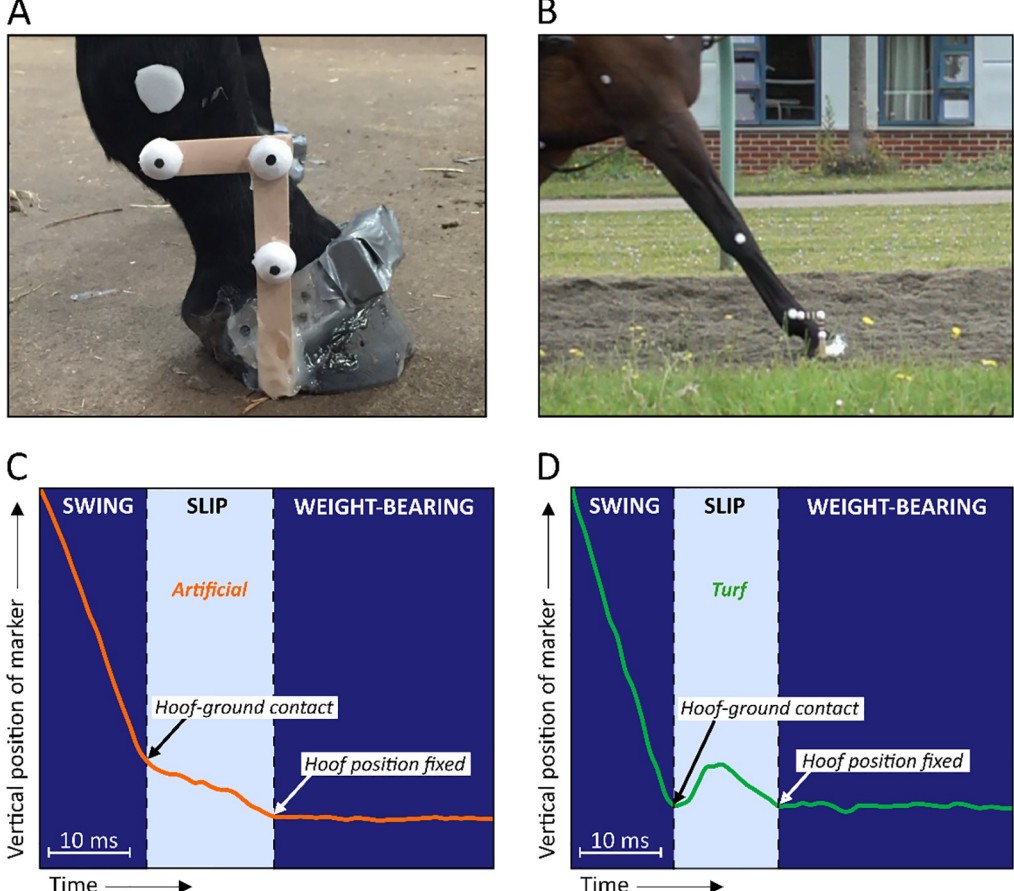

**Fig 1. Illustration of the change in vertical position of a marker fixed on the lateral aspect of the hoof. A)**
Photograph of marker wand on the lateral aspect of the hoof. The accelerometer visible on the dorsal hoof wall was used
in a different study component [26]. **B)** Screenshot of hoof at initial ground contact from Tracker software. C) Typical
vertical trajectory of the marker into soft artificial surface during slip phase. **D)** Typical vertical trajectory of the marker
on turf, incorporating bump over surface after the initial contact.

view the video footage, alongside plots of tracked x (horizontal) and y (vertical) coordinates to
quantify a best estimate of the entire slip/sink window when the hoof was moving. The num-
ber of frames taken to complete the slip/sink phase was noted and used to calculate 'slip' dura-
tion (in frames).

To account for a possible influence of gallop speed on breakover duration, the mean gallop
speed recorded by the GPS devices between the start and end of the camera set-up was evalu-
ated. As detailed in [41], this was achieved by first identifying the location of the cameras using
satellite imagery: they were identified to fall between 52.26579 N, 0.414454 E and 52.26564 N,
0.414711 E on the artificial track, and between 52.2657 N, 0.414237 E and 52.26556 N,
0.414531 E on the turf track. The speed and position of the horse in latitude-longitude space
was then plotted alongside the camera position to identify the relevant speed data.

## Statistics

Linear mixed models were implemented in SPSS to test for significant differences in hoof slip
at landing, under the different shoe and surface conditions. Shoe, surface, speed, "shoe*surface

interaction", "shoe*speed interaction" and "surface*speed interaction" were defined as fixed factors, and horse—jockey pair ID and day were defined as random factors. Speed was also included as a covariate. The p value outputs for the interaction terms of these initial linear mixed models were evaluated. If any p values for interaction terms exceeded 0.1, then these terms were removed so 'final' models could be run with a reduced number of fixed terms to lower statistical noise. Histograms of models' residuals were plotted, and normality was confirmed. The significance threshold in all statistical tests was set at $p \leq 0.05$.

## Results

Table 1 presents a summary of the raw data for slip duration data sub-divided by shoe—surface combination and limb. The mean speed per condition is also indicated. The raw slip data are summarised according to surface and shoeing condition effects in Figs 2 and 3, respectively. Combined shoe and surface effects are shown in Fig 4. The data from the linear mixed models are summarised below for each limb type and reported in Tables 2–5.

### Non-leading hindlimb

Preliminary models for the non-leading hindlimb indicated that all interaction terms had p values $\geq 0.245$. The final model indicated that shoeing condition and speed had insignificant effects on hoof slip duration (p = 0.826 and p = 0.595, respectively) but surface was significant (p = 0.018). The estimated marginal means for surface effects indicated that slip duration was 3.2 ms longer on turf than on the artificial surface.

### Leading hindlimb

Preliminary models for the leading hindlimb indicated that all interaction terms had p values $\geq 0.117$. The final model indicated that speed did not have a significant effect on slip duration (p = 0.225) but shoeing condition and surface had a significant effect on hoof slip duration (p = 0.044 and p = 0.025, respectively). The estimated marginal means indicated that slip durations were 3.7 ms longer for the aluminium shoe than for the barefoot condition, and 2.5 ms longer on the artificial surface than on turf. Although the models did not identify a significant relationship between slip duration and speed, when the raw data were plotted (Fig 5) there appeared to be a weak positive correlation (p = 0.031, $r^2$ = 0.053).

### Non-leading forelimb

Preliminary models for the non-leading forelimb indicated that all interaction terms had p values $\geq 0.127$. The final model indicated that shoeing condition and speed did not have a significant effect on slip duration (p = 0.656, and p = 0.515, respectively), but surface was significant. The estimated marginal means for surface effects indicated that slip duration was 3.5 ms longer on turf.

### Leading forelimb

Preliminary models for the leading forelimb indicated that all interaction terms had p values $\geq 0.233$. The final model indicated that shoeing condition had no significant effect (p = 0.249) but surface was significant (p = 0.050). The estimated marginal means for surface effects suggested that slip duration was 2.1 ms longer on turf. The model also indicated that speed had a significant effect on hoof slip in the leading forelimb (p = 0.001). There was a decreasing slip duration with increasing speed (Fig 2, $r^2$ = 0.169).

**Table 1. Summary of slip duration and speed data sub-divided by shoe–surface combination and limb.** The number of horse–jockey pairs available in the analysis of each condition is stated.

| Shoe | Surface | Limb | Number of observations | Number of horse-jockey pairs | Mean slip duration (ms) | 2 S.D. slip duration (ms) | Mean speed (km h⁻¹) | 2 S.D. speed (km h⁻¹) |
|---|---|---|---|---|---|---|---|---|
| Aluminium | Artificial | Leading forelimb | 15 | 12 | 19.53 | 8.31 | 42.94 | 12.71 |
| Aluminium | Artificial | Non-leading forelimb | 17 | 12 | 18.82 | 9.20 | 43.30 | 11.58 |
| Aluminium | Artificial | Leading hindlimb | 14 | 11 | 22.29 | 11.19 | 43.20 | 13.02 |
| Aluminium | Artificial | Non-leading hindlimb | 17 | 12 | 18.94 | 9.23 | 42.95 | 11.13 |
| Aluminium | Turf | Leading forelimb | 8 | 7 | 23.13 | 9.16 | 35.91 | 7.54 |
| Aluminium | Turf | Non-leading forelimb | 14 | 7 | 24.21 | 11.40 | 37.59 | 7.80 |
| Aluminium | Turf | Leading hindlimb | 8 | 7 | 17.88 | 11.73 | 35.91 | 7.54 |
| Aluminium | Turf | Non-leading hindlimb | 14 | 7 | 21.79 | 14.47 | 37.59 | 7.80 |
| Barefoot | Artificial | Leading forelimb | 14 | 13 | 20.29 | 11.70 | 41.30 | 15.24 |
| Barefoot | Artificial | Non-leading forelimb | 19 | 14 | 19.47 | 10.36 | 42.92 | 14.27 |
| Barefoot | Artificial | Leading hindlimb | 14 | 12 | 18.86 | 8.90 | 42.65 | 14.96 |
| Barefoot | Artificial | Non-leading hindlimb | 17 | 13 | 18.82 | 10.15 | 41.99 | 13.91 |
| Barefoot | Turf | Leading forelimb | 11 | 9 | 20.91 | 11.75 | 37.14 | 11.77 |
| Barefoot | Turf | Non-leading forelimb | 9 | 8 | 19.89 | 8.51 | 38.87 | 13.97 |
| Barefoot | Turf | Leading hindlimb | 12 | 9 | 15.50 | 6.47 | 38.72 | 15.69 |
| Barefoot | Turf | Non-leading hindlimb | 9 | 8 | 20.67 | 16.64 | 38.87 | 13.97 |
| GluShu | Artificial | Leading forelimb | 9 | 9 | 19.89 | 8.63 | 38.28 | 10.08 |
| GluShu | Artificial | Non-leading forelimb | 13 | 10 | 19.54 | 9.00 | 38.15 | 9.32 |
| GluShu | Artificial | Leading hindlimb | 9 | 9 | 17.56 | 7.56 | 38.28 | 10.08 |
| GluShu | Artificial | Non-leading hindlimb | 13 | 10 | 20.62 | 13.28 | 38.15 | 9.32 |
| GluShu | Turf | Leading forelimb | 13 | 8 | 24.69 | 13.62 | 35.09 | 11.79 |
| GluShu | Turf | Non-leading forelimb | 10 | 7 | 21.70 | 13.33 | 37.14 | 8.05 |
| GluShu | Turf | Leading hindlimb | 11 | 8 | 15.73 | 7.75 | 35.97 | 11.62 |
| GluShu | Turf | Non-leading hindlimb | 11 | 7 | 22.73 | 13.77 | 37.94 | 10.16 |
| Steel | Artificial | Leading forelimb | 13 | 11 | 21.00 | 6.43 | 41.71 | 13.91 |
| Steel | Artificial | Non-leading forelimb | 12 | 10 | 18.75 | 7.73 | 42.41 | 12.74 |
| Steel | Artificial | Leading hindlimb | 11 | 9 | 20.18 | 8.71 | 40.87 | 12.80 |
| Steel | Artificial | Non-leading hindlimb | 11 | 9 | 18.55 | 10.17 | 40.69 | 11.93 |
| Steel | Turf | Leading forelimb | 10 | 7 | 23.70 | 7.49 | 41.25 | 13.52 |

*(Continued)*

**Table 1.** (Continued)

| Shoe | Surface | Limb | Number of observations | Number of horse-jockey pairs | Mean slip duration (ms) | 2 S.D. slip duration (ms) | Mean speed (km h$^{-1}$) | 2 S.D. speed (km h$^{-1}$) |
|---|---|---|---|---|---|---|---|---|
| Steel | Turf | Non-leading forelimb | 13 | 9 | 24.92 | 11.87 | 38.77 | 11.36 |
| Steel | Turf | Leading hindlimb | 9 | 7 | 16.89 | 12.39 | 41.33 | 14.33 |
| Steel | Turf | Non-leading hindlimb | 9 | 8 | 23.11 | 9.97 | 40.53 | 10.79 |

## Discussion

The impact of shoe and surface conditions on hoof slip duration at gallop depended on the limb evaluated. Surface type significantly affected hoof slip in all limbs, and for the forelimbs and the non-leading hindlimb hoof slip duration was longer on turf compared to the artificial surface, by 2.1 to 3.5 ms. However, the leading hindlimb was associated with a mean slip duration that was 2.5 ms shorter on turf compared to on the artificial surface. In addition, the leading hindlimb was the only limb that was sensitive to shoeing condition, with a significantly longer slip duration associated with the aluminium shoe compared to barefoot.

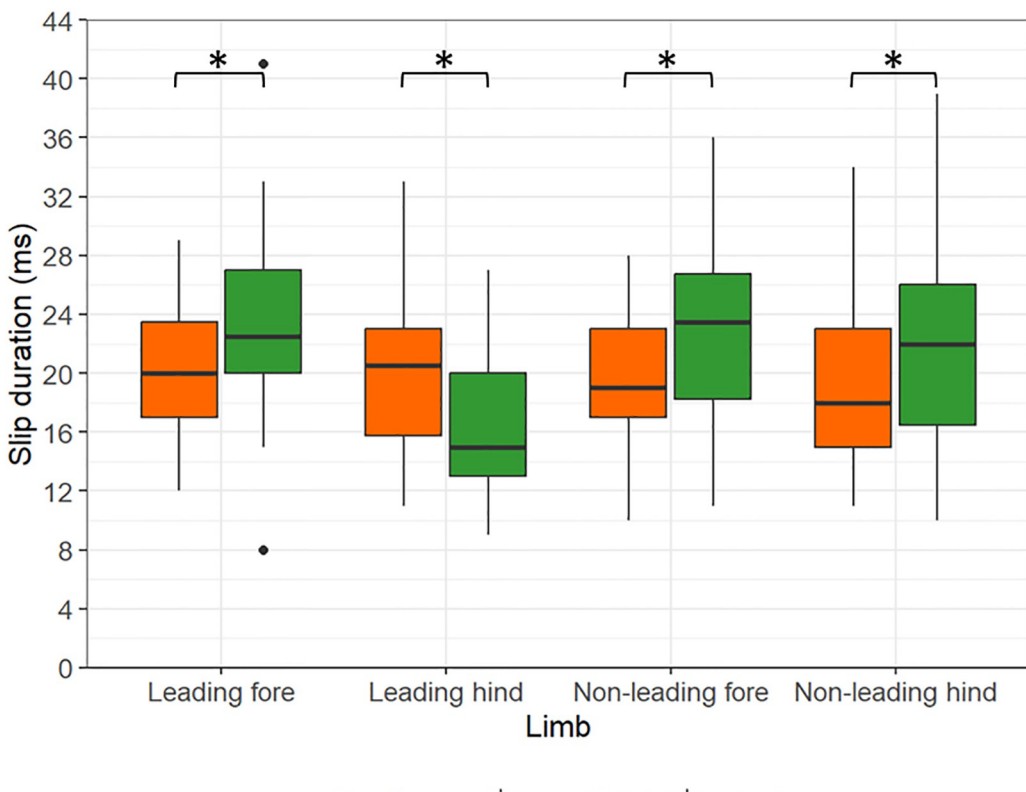

**Fig 2. Boxplots illustrating the influence of surface on hoof slip duration for each limb.** Data for the artificial surface are shown in orange and data for the turf surface are shown in green. All comparisons were significant (*). The two outliers for the leading forelimb were from different horses.

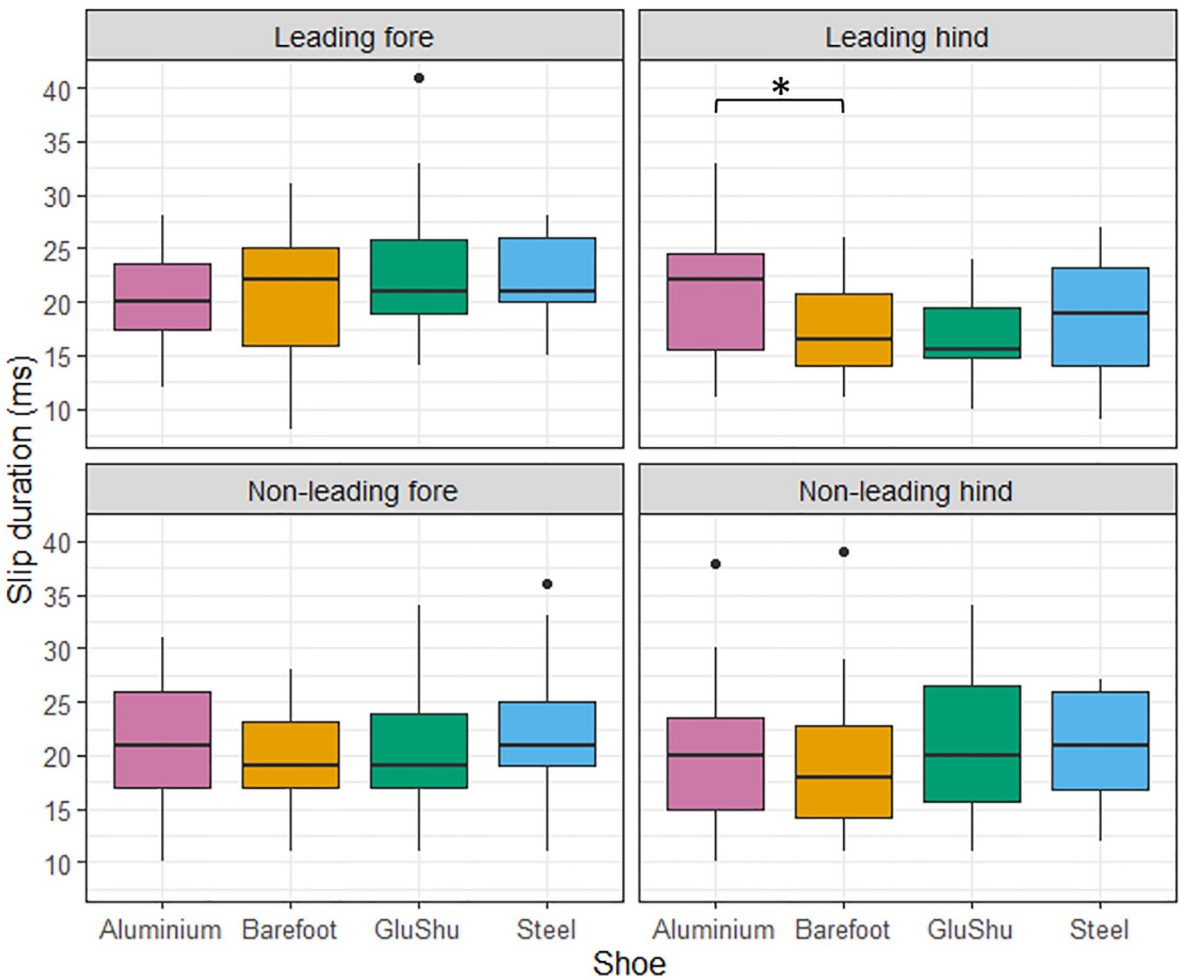

**Fig 3. Boxplots illustrating the influence of shoeing condition on hoof slip duration for each limb.** Data for the aluminium shoes are shown in pink; data for barefoot shoeing condition are shown in yellow; data for the GluShu shoes are shown in green; and data for the steel shoes are shown in blue. The significant difference between the aluminium and barefoot condition in the leading hindlimb is highlighted (*). Please note that the four outliers indicated came from four different horses.

The reason for the differing response of the leading hindlimb may be linked to its key role in diverting the centre of mass trajectory from downwards and forwards to upwards and forwards during the stride cycle [43]. Overall, slip durations were also shortest in the leading hindlimb, but similar amongst the other limb types (Table 1). In asymmetrical gaits, including canter and gallop, the leading hindlimb reaches out further ahead of the body during the swing phase and is more protracted compared to the non-leading hindlimb [44, 45]. The leading hindlimb reaches a greater distance in a given swing time by having more flexed elbow, hip and tarsal joints [45]. Coupled with this, previous work has also indicated that the leading hindlimb has the highest vertical hoof velocities, and reduced horizontal velocities relative to the non-leading hindlimb [33]. This should minimise the delay in force redirection and increase grip for acceleration and manoeuvring using the hindlimb musculature, which will be particularly important on high-speed turns [46]. A reduced slip duration should mean that there is increased time for the hoof to produce vertical force efficiently, and this may limit peak force. Force plate data from galloping horses are in support of this idea, as peak ground reaction

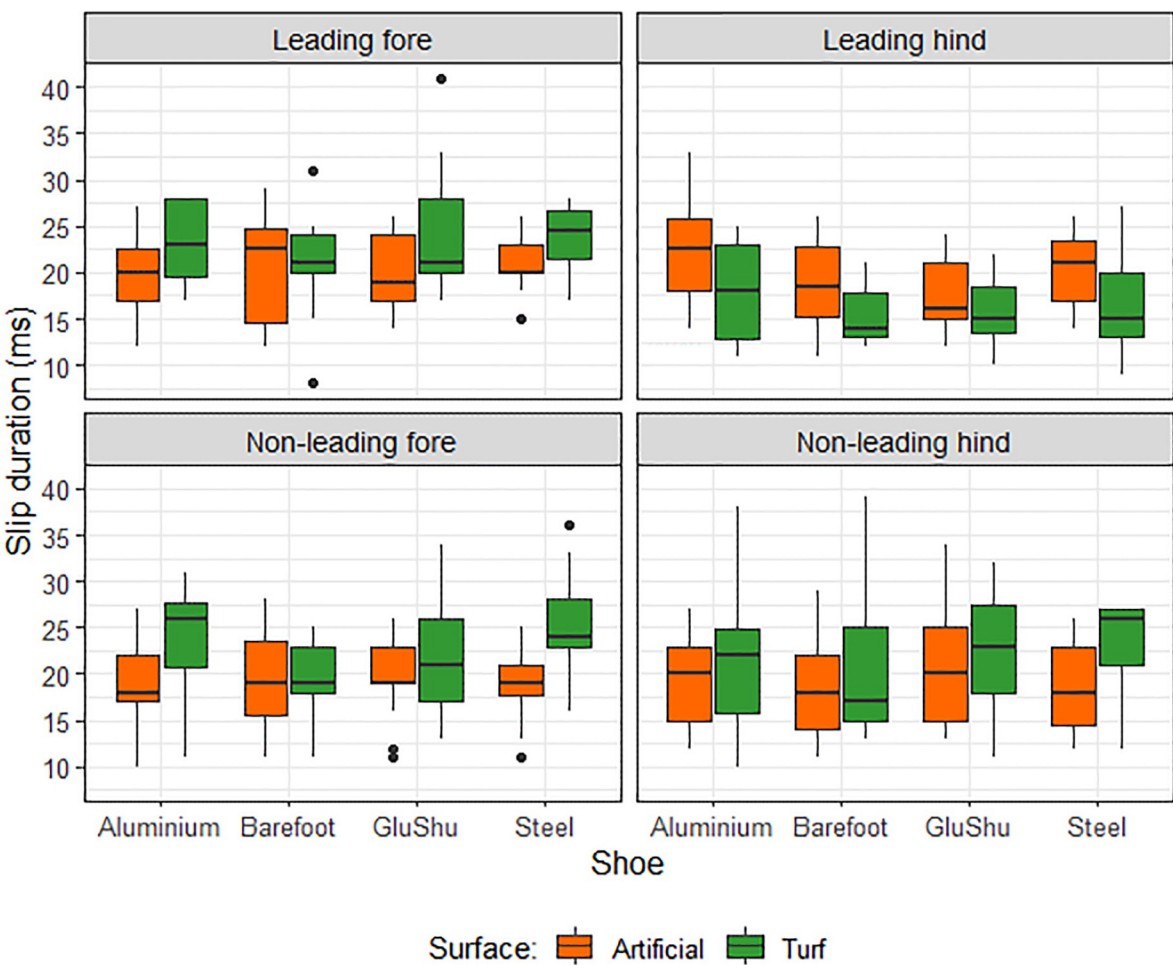

**Fig 4. Boxplots illustrating the influence of surface and shoeing condition on slip duration for each limb.** Data for the artificial surface are shown in orange and data for the turf surface are shown in green.

**Table 2. Statistical results for the effect of shoe, surface and speed on hoof slip duration in each limb type.** Data are from the linear mixed models.

| Limb | Source | F | Sig. |
|---|---|---|---|
| Leading forelimb | Shoe | 1.40 | 0.249 |
| | Surface | 3.95 | 0.050 |
| | Speed | 11.51 | 0.001 |
| Non-leading forelimb | Shoe | 0.54 | 0.656 |
| | Surface | 10.37 | 0.002 |
| | Speed | 0.43 | 0.515 |
| Leading hindlimb | Shoe | 2.82 | 0.044 |
| | Surface | 5.22 | 0.025 |
| | Speed | 1.49 | 0.225 |
| Non-leading hindlimb | Shoe | 0.30 | 0.826 |
| | Surface | 5.96 | 0.018 |
| | Speed | 0.29 | 0.595 |

**Table 3. Linear mixed model estimated marginal means for surface effects on hoof slip duration.**

| Limb | Surface | Mean | Std. Error | 95% Confidence Interval (lower bound) | 95% Confidence Interval (upper bound) |
|---|---|---|---|---|---|
| Leading forelimb | Artificial | 20.64 | 0.82 | 19.01 | 22.26 |
| | Turf | 22.71 | 0.89 | 20.94 | 24.49 |
| Non-leading forelimb | Artificial | 19.32 | 0.68 | 17.97 | 20.68 |
| | Turf | 22.79 | 0.79 | 21.21 | 24.36 |
| Leading hindlimb | Artificial | 19.45 | 0.86 | 17.74 | 21.17 |
| | Turf | 17.00 | 0.93 | 15.14 | 18.86 |
| Non-leading hindlimb | Artificial | 19.06 | 1.03 | 16.88 | 21.23 |
| | Turf | 22.22 | 1.18 | 19.76 | 24.69 |

**Table 4. Linear mixed model estimated marginal means for shoeing condition effects on hoof slip duration.**

| Limb | Shoeing condition | Mean | Std. Error | 95% Confidence Interval (lower bound) | 95% Confidence Interval (upper bound) |
|---|---|---|---|---|---|
| Leading forelimb | Aluminium | 21.32 | 1.06 | 19.21 | 23.42 |
| | Barefoot | 20.40 | 1.01 | 18.39 | 22.41 |
| | GluShu | 22.00 | 1.12 | 19.77 | 24.22 |
| | Steel | 22.99 | 1.08 | 20.85 | 25.13 |
| Non-leading forelimb | Aluminium | 21.45 | 0.93 | 19.60 | 23.30 |
| | Barefoot | 20.31 | 1.00 | 18.33 | 22.28 |
| | GluShu | 20.56 | 1.11 | 18.35 | 22.76 |
| | Steel | 21.91 | 1.04 | 19.85 | 23.96 |
| Leading hindlimb | Aluminium | 20.15 | 1.09 | 17.98 | 22.32 |
| | Barefoot | 16.49 | 1.02 | 14.47 | 18.52 |
| | GluShu | 18.32 | 1.20 | 15.94 | 20.70 |
| | Steel | 17.94 | 1.14 | 15.67 | 20.21 |
| Non-leading hindlimb | Aluminium | 20.32 | 1.28 | 17.69 | 22.94 |
| | Barefoot | 20.04 | 1.33 | 17.35 | 22.73 |
| | GluShu | 21.63 | 1.41 | 18.77 | 24.48 |
| | Steel | 20.58 | 1.48 | 17.61 | 23.55 |

forces were found to be lowest in the leading hindlimb [47]. Consequently, minimising slip duration during propulsive efforts in the hind end may also lessen the risk of injury.

In terms of our surface observations, if there is a higher vertical hoof velocity in the leading hindlimb then more rapid vertical sinking into the ground surface should be expected, when surface properties permit. Soft deformable surfaces, such as all-weather waxed surfaces, give rise to higher vertical hoof velocities, when compared to turf [42, 48, 49]. Hence, in this study, the soft artificial surface should facilitate proportionally more 'vertical sink' in the slip phase for the leading hindlimb than for the other limbs. In contrast, if the hooves of non-leading hindlimb and the forelimbs experience higher horizontal hoof velocities [33], then a longer horizontal sliding component of the total slip period may be expected on the turf surface in these limbs. This is because there should be reduced resistance to the forwards movement of these limbs as they will be less anchored into the less compliant turf surface, where vertical hoof sink is limited. Turf is also expected to have a lower coefficient of static friction, which will allow the hoof to slide more easily, thereby increasing hoof deceleration time and distance [36]. In the forelimbs and non-leading hind, we found slip durations were 14–19% lower on the artificial surface than on turf. For comparison, a study investigating hoof slip distances on dirt

**Table 5. Linear mixed model estimated marginal means for shoe-surface effects on hoof slip duration.**

| Limb | Shoeing condition | Surface | Mean | Std. Error | 95% Confidence Interval (lower bound) | 95% Confidence Interval (upper bound) |
|------|-------------------|---------|------|-----------|----------------------------------------|----------------------------------------|
| Leading forelimb | Aluminium | Artificial | 20.28 | 1.13 | 18.03 | 22.53 |
| | | Turf | 22.35 | 1.23 | 19.92 | 24.79 |
| | Barefoot | Artificial | 19.36 | 1.10 | 17.16 | 21.55 |
| | | Turf | 21.44 | 1.17 | 19.11 | 23.76 |
| | GluShu | Artificial | 20.96 | 1.22 | 18.53 | 23.38 |
| | | Turf | 23.03 | 1.25 | 20.54 | 25.52 |
| | Steel | Artificial | 21.95 | 1.18 | 19.60 | 24.30 |
| | | Turf | 24.03 | 1.21 | 21.63 | 26.43 |
| Non-leading forelimb | Aluminium | Artificial | 19.72 | 1.06 | 17.62 | 21.82 |
| | | Turf | 23.19 | 1.10 | 21.01 | 25.36 |
| | Barefoot | Artificial | 18.57 | 1.06 | 16.47 | 20.67 |
| | | Turf | 22.04 | 1.20 | 19.65 | 24.43 |
| | GluShu | Artificial | 18.82 | 1.17 | 16.50 | 21.14 |
| | | Turf | 22.29 | 1.29 | 19.72 | 24.86 |
| | Steel | Artificial | 20.17 | 1.18 | 17.83 | 22.52 |
| | | Turf | 23.64 | 1.15 | 21.35 | 25.93 |
| Leading hindlimb | Aluminium | Artificial | 21.38 | 1.17 | 19.05 | 23.71 |
| | | Turf | 18.93 | 1.26 | 16.42 | 21.43 |
| | Barefoot | Artificial | 17.72 | 1.14 | 15.46 | 19.98 |
| | | Turf | 15.27 | 1.17 | 12.95 | 17.59 |
| | GluShu | Artificial | 19.55 | 1.28 | 17.01 | 22.08 |
| | | Turf | 17.10 | 1.35 | 14.41 | 19.78 |
| | Steel | Artificial | 19.17 | 1.25 | 16.68 | 21.66 |
| | | Turf | 16.72 | 1.27 | 14.18 | 19.25 |
| Non-leading hindlimb | Aluminium | Artificial | 18.73 | 1.39 | 15.92 | 21.54 |
| | | Turf | 21.90 | 1.49 | 18.89 | 24.91 |
| | Barefoot | Artificial | 18.46 | 1.39 | 15.65 | 21.26 |
| | | Turf | 21.62 | 1.57 | 18.48 | 24.77 |
| | GluShu | Artificial | 20.04 | 1.51 | 17.00 | 23.08 |
| | | Turf | 23.21 | 1.60 | 19.99 | 26.43 |
| | Steel | Artificial | 18.99 | 1.57 | 15.86 | 22.13 |
| | | Turf | 22.16 | 1.66 | 18.83 | 25.49 |

versus synthetic surfaces in horses breezing found their synthetic surface was associated with 40% less horizontal translation [24]. However, it is not clear to what extent these studies considered the vertical versus horizontal movement of the hooves separately at landing. Our study has emphasised the differing response of hooves on different surfaces depending on the associated limb, and this is likely a product of altered horizontal and vertical hoof velocities amongst the different hooves in the galloping gait. Therefore, when considering the impact of different racetrack properties for hoof landing kinematics, racehorse trainers and associated personnel will need to evaluate each limb/hoof separately. In our study, jockeys perceived there to be increased slip on the turf than on the artificial track [39], suggesting their perceptions of the hoof-ground interaction during landing most closely align with the behaviour of the forelimbs and non-leading hindlimb. However, it is also important to note that specific surface properties, including temperature, moisture content and composition will affect surface response at hoof-contact [50, 51], and may influence slip durations. All the outliers indicated on Figs 2

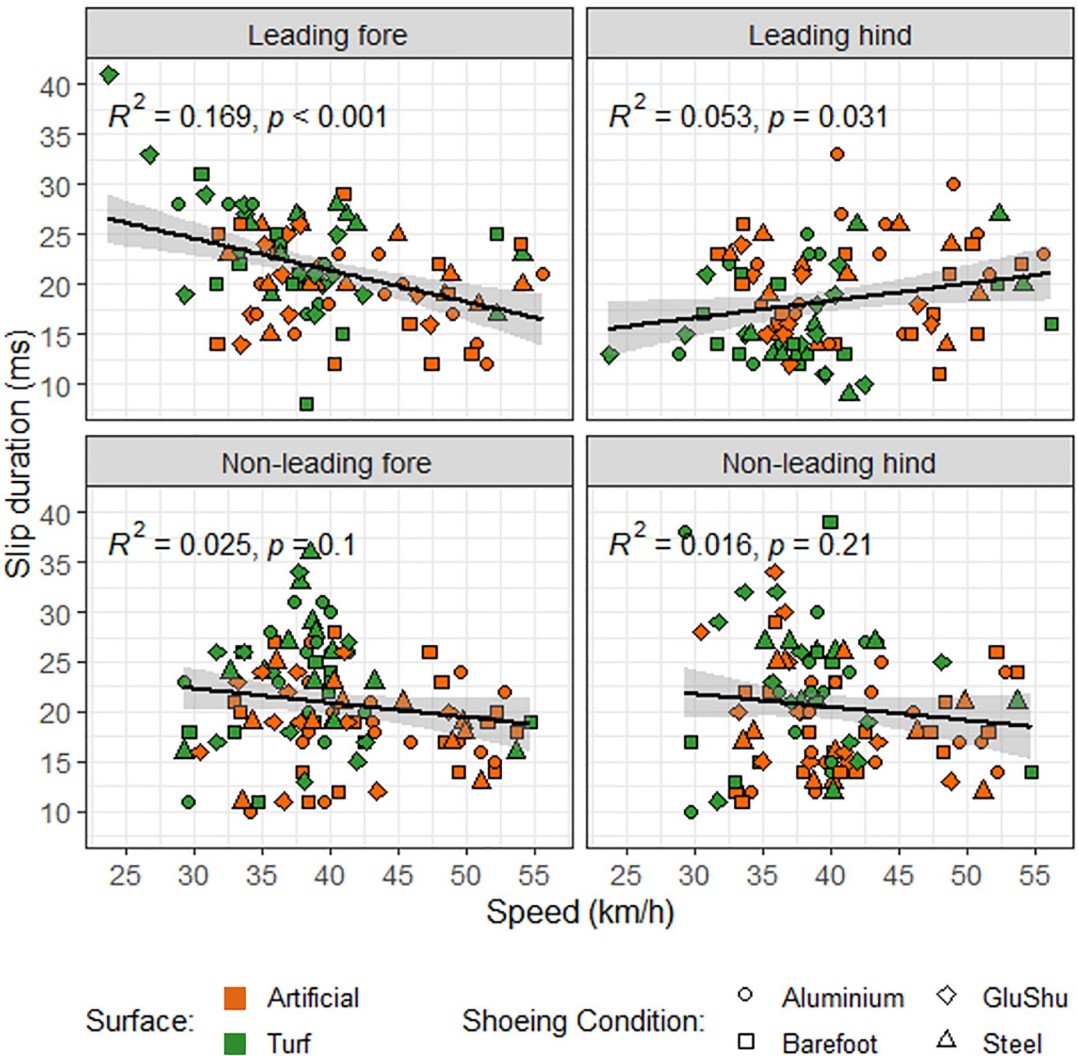

**Fig 5. Relationship between slip duration and speed in the leading forelimb.** The solid black line represents the linear best fit to the raw data with the 95% confidence interval shown as a grey band. $R^2$ and p values indicated were quantified based on the raw data plotted (rather than linear mixed model outputs). Data are coloured according to surface and shapes indicate shoeing condition.

and 3 represent data collected on turf surfaces, suggesting the turf was more prone to extreme variations in hoof slip. It is a limitation of this study that surface properties were not objectively tested.

The shorter slip durations for the leading hindlimb are consistent with higher impact accelerations in this limb [26]. Peak hoof decelerations have been found to correlate with peak hoof ground reaction force in horses galloping horses on dirt, synthetic and turf surfaces [25]. Therefore, shorter slip durations and faster hoof decelerations in the leading hindlimb are expected to be associated with higher limb loading and greater stresses being transferred to the proximal musculoskeletal structures, which may increase injury risk [24, 34]. For example, repetitive impulse loading can damage subchondral bone and articular cartilage [13, 15]. This effect may be exacerbated on the firmer turf, where tri-axial impact accelerations are reported to be higher compared to the softer artificial surface [26]. However, it is interesting to note

that there appears to be a lower incidence of injuries in hindlimbs [22], suggesting that other factors could be important in the aetiology of injuries. At present, differences in reported incidences of musculoskeletal injuries and common and conflicting risk factors across different racetracks and countries [52] make it challenging to identify the limb(s) most at risk from alterations to hoof slip. Also, it is worth emphasising that establishing an appropriate balance between vertical sink and horizontal slide is important, and limitations to overall 'slip' durations are required. For example, in jumping horses greater slide increases extension of the coffin joint and therefore the load on the deep digital flexor tendons, and a greater penetration depth of the toe of the hoof is expected to increase risk of injury to the collateral ligament of the coffin joint [53]. In addition, it appears that there is not necessarily a simple horizontal translation of the hoof across certain surfaces, such as turf, but often a 'bounce' after the initial contact (Fig 1D). This could indicate that a lot of the impact shock is absorbed by the structures of the hoof and distal limb, rather than by the deformation of the surface, which may predispose to injuries such as sore shins, fractures to the cannon bone, splints or tendon injuries.

Shoeing has been proposed as an important factor for dissipating foot impact forces [2, 54]. In the current study, there were few differences amongst slip durations in the different shoeing conditions, with only a significantly longer slip duration being found for the aluminium shoe condition when compared to barefoot in the leading hindlimb (Fig 3). This goes against jockey opinion, which indicated that there were several significant differences in slip amongst the shoeing conditions, and the jockeys actually suggested that slip was decreased for the aluminium shoe compared to barefoot [39]. However, we did not ask the jockeys to differentiate between slip duration and slip distance, which may explain the discrepancy. The reason for the observed increase in slip durations for the aluminium shoe versus the barefoot condition, as quantified in this study, may be linked to differing limb trajectory during the swing phase and/or just before landing, plus the relatively low mass of the aluminium shoe (relative to the other shoe types) serving to prolong the time taken for the hoof to stabilise and sink during landing. Nonetheless, the general similarity between slip times in different shoeing conditions in this study and previous work [2, 35], could suggest that horses alter their gait to compensate for grip characteristics of the shoe and maintain a constant slip time. For example, the hoof slip duration in cantering horses trialling shoes with and without a lateral heel stud, found that slip durations were only affected in the non-leading forelimb [35]. In addition, slip times and distances were not significantly different for horses trotting over concrete in either steel, rubber or plastic shoes, despite the craniocaudal decelerative force being reduced in the plastic shoes [2]. As in humans, it is possible that under slippery shoe-surface conditions, awareness of a potential slip could alter how the different limbs approach the ground surface and prior slip experience may alter the anticipatory muscle activation and how the hoof interacts with the floor [55]. A horse may alter its limb flight patterns and foot placement to compensate for different shoeing conditions through altering joint angles, joint angular velocities and foot velocity at impact [56].

The mean slip duration recorded here was 20.3 ± 11.2 ms (mean ± 2 s.d., unless otherwise stated), with values ranging from 8–41 ms. These data appear consistent with values that may be calculated from foot velocity and slip distance plots [33] and are mostly within error of hoof slip durations of 37 ± 14 ms and 31 ± 14 ms previously quantified on dirt and synthetic surfaces, respectively, at gallop [24]; these studies also quantified slip from video footage. For horses trotting on concrete, slip durations were approximately 20 ms, and hence also similar to the values we recorded in horses galloping on softer turf and artificial surfaces. Hoof slip durations quantified at slower gaits were also of similar magnitude; for example, horses trotting on sand experienced a mean total slip time of 28.1 ± 8.8 ms [1] and on a stone dust track, the absolute length of the hoof-braking period for Standardbred trotters was between 30 and

50 ms, independent of trot speed [34]. Mean slip durations amongst limbs in cantering horses have been reported to range from 30–39 ms [35]. However, direct comparisons of gait and shoe-surface effects on slip duration across studies are made challenging by different data collection and analysis methods, and the inclusion of different horse types that contrast in conformation and discipline, for example. Using a purely kinematic technique to detect hoof contact and slip-stop may introduce errors, particularly at low frame rates [1, 33, 35], and the above data indicate quite high variability on measurements. It is also worth noting that our models identified no significant effect of speed on hoof slip duration for the non-leading forelimb and hindlimbs, and there was only a weak negative correlation for the leading forelimb (Fig 5). The observation of a reduction in slip duration in the leading forelimb as speed increases may be related to the fact that at higher speeds there is less time available for the hoof to be in contact with the ground. However, given that the correlation is weak, the true impact of this relationship is likely to be minor. Instead, it seems that the specific hoof in question and the surface involved have a greater influence on total slip duration than overall speed.

Improving understanding of the factors controlling slip duration is important, as the slip phase represents a period of uncertainty for the neuromechanics of the horse and a period during which force redirection is delayed. At present, we do not know the most clinically relevant limb in which to prioritise optimal slip type. If forelimbs are most likely to fracture [22], the emphasis might initially be placed on further investigations into the forelimbs. However, the exact nuances of the likelihood of injury are a complex interplay of various additional race characteristics including, but not limited to, horse age, sex, race distance and field size [18, 19, 52]. In addition, alterations to traction at the hoof-surface interface can also impact upper body movement asymmetry [57], and the relevance of this in racing contexts requires investigation. Future work should seek to quantify the implications of altered slip durations on racehorse upper body biomechanics at gallop, as this will be relevant for injury mechanics in both the horses and their jockeys.

## Conclusion

This study investigated the duration of hoof slip in galloping racehorses as they trialled eight shoe-surface combinations. We found that hoof slip duration was limb specific: the forelimbs and the non-leading hind had longer hoof slip durations on turf compared to the artificial surface, whereas the leading limb had shorter hoof slip durations on turf. The leading hindlimb was also sensitive to shoeing condition, with increased slip durations found for an aluminium shoe compared to barefoot. A differing response of the leading limb to shoe and surface conditions, and its overall shorter hoof slip durations, may be related to its important role in redirecting the horse's centre of mass during the stride cycle and its higher vertical hoof velocities pre-impact. The interaction between hooves and the surfaces they are galloping over is at the heart of the risk of slippage, fractures and falls. Therefore, these findings are relevant for understanding the stability of the hoof and distal limb during landing and the likely resulting concussive forces and loading rates, which may bear relevance for injury risk.

## Acknowledgments

The authors would like to thank the British Racing School for facilitating access to horses, jockeys and facilities. Jessica Josephson, Edward Evans, Alice Morrell, Morgan Ruble and Hazel Birch-Ellis from the Royal Veterinary College and Simon Curtis are all thanked for their assistance and support with data collection.

## Author Contributions

**Conceptualization:** Kate Horan, Renate Weller, Thilo Pfau.

**Data curation:** Kate Horan.

**Formal analysis:** Kate Horan.

**Funding acquisition:** Renate Weller, Thilo Pfau.

**Investigation:** Kate Horan, James Coburn, Kieran Kourdache, Peter Day, Henry Carnall, Liam Brinkley, Dan Harborne, Lucy Hammond, Sean Millard.

**Methodology:** Kate Horan.

**Project administration:** Kate Horan, Thilo Pfau.

**Supervision:** Thilo Pfau.

**Validation:** Kate Horan.

**Visualization:** Kate Horan.

**Writing – original draft:** Kate Horan.

**Writing – review & editing:** Kate Horan, Thilo Pfau.

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
