## [Decision Letter · Decision Letter 0]

19 Jun 2024

PONE-D-24-21823Hoof slip duration at impact in galloping Thoroughbred ex-racehorses trialling eight shoe-surface combinationsPLOS ONE

Dear Dr. Horan,

Thank you for submitting your manuscript to PLOS ONE. After careful consideration, we feel that it has merit but does not fully meet PLOS ONE’s publication criteria as it currently stands. Therefore, we invite you to submit a revised version of the manuscript that addresses the points raised during the review process.

We look forward to receiving your revised manuscript.

Kind regards,

Tzen-Yuh Chiang

Academic Editor

PLOS ONE

Journal Requirements:

"This research was funded by the Horserace Betting Levy Board, project 4497, Prj786, grant titled ‘S.A.F.E.R.’ (Shoe Assessment for Equine Racing)."

4. Please note that funding information should not appear in the Acknowledgments section or other areas of your manuscript. We will only publish funding information present in the Funding Statement section of the online submission form. Please remove any funding-related text from the manuscript. 

5. Thank you for stating the following in the Competing Interest section: 

"I have read the journal's policy and the authors of this manuscript have the following competing interests: J.C. owns the company James Coburn AWCF Farriers Ltd., which employed H.C., L.B. and D.H. at the time of the study. J.C., P.D., H.C., L.B. and D.H. and are now registered farriers. T.P. and R.W. are the owners of EquiGait, a provider of gait analysis products and services. This does not alter our adherence to all polices on sharing data and materials. The funders had no role in the design of the study or the collection, analyses and interpretation of data."

We note that one or more of the authors are employed by a commercial company: James Coburn AWCF Farriers Ltd.

2) Please also provide an updated Competing Interests Statement declaring this commercial affiliation along with any other relevant declarations relating to employment, consultancy, patents, products in development, or marketed products, etc.  

Within your Competing Interests Statement, please confirm that this commercial affiliation does not alter your adherence to all PLOS ONE policies on sharing data and materials by including the following statement: ""This does not alter our adherence to  PLOS ONE policies on sharing data and materials.” (as detailed online in our guide for authors http://journals.plos.org/plosone/s/competing-interests) . If this adherence statement is not accurate and  there are restrictions on sharing of data and/or materials, please state these. Please note that we cannot proceed with consideration of your article until this information has been declared.

Reviewers' comments:

Reviewer's Responses to Questions

**Comments to the Author**

1. Is the manuscript technically sound, and do the data support the conclusions?

Reviewer #1: Partly

Reviewer #2: Yes

Reviewer #3: Yes

2. Has the statistical analysis been performed appropriately and rigorously? 

Reviewer #1: I Don't Know

Reviewer #2: Yes

Reviewer #3: Yes

3. Have the authors made all data underlying the findings in their manuscript fully available?

Reviewer #1: Yes

Reviewer #2: Yes

Reviewer #3: Yes

4. Is the manuscript presented in an intelligible fashion and written in standard English?

Reviewer #1: Yes

Reviewer #2: Yes

Reviewer #3: Yes

5. Review Comments to the Author

Reviewer #1: Overall this paper provides some data on the slip in different limbs during gallop with 4 different shoeing conditions on 2 different surfaces. These observations are of potential use to underpin investigations into the development of injury in sport horses.

The measurement of hoof slip through kinematics is prone to many errors that were not mentioned or discussed although some of the causes of loss of data were mentioned in lines 120-123.

As one horse was ridden by 2 jockeys, the number of jockey/horse pairs was given as 14 for the 13 horses. It is not clear to this reviewer how this was treated in the statistics, nor whether that one horse provided data for both jockeys in all the different treatments.

It seems that not all the 13 horses that were used provided data for all the different conditions. Table 1 shows that the number of observations of hoof slip varied between 7 and 19 with more than half (17) below the number of horses for any of the 32 combinations of 4 limbs, 2 surfaces and 4 shoeing types. Hence it is clear that not all the 13 horses provided data in all the 32 possible combinations for slip in each of the limbs.

Further, in Table 1 the number of horse jockey pairs that provided data for each of the 8 experimental conditions is below 13 in all but one condition (barefoot artificial).

The authors are to be congratulated for clearly providing this data, but it does require some further discussion as to whether the surface characteristics (limitation mentioned in line 254) that were described as varying from soft to good/firm (line99) for the grass showed any relationship with the likelihood of a measurement being acquired. This might then better support the statement in Line 308 “However, overall it seems that the specific hoof in question and the surface involved have a greater influence on total slip duration than overall speed.”

Reviewer #2: Congratulations on the series of articles. My minor revision decision implies the possible correction of the following aspects:

L 83 I suggest using the International system and converting horse hand to meters.

L 108-111 and L137 there is a discrepancy between the text (A. Photograph of marker wand on the hoof.) and the additional presence in the photograph of an accelerometer mounted to the hoof.

L 223 “hinlimb” is misspelled

L 222-224 For a better accuracy and understanding, I suggest completing the cited information from Back W. et al (reference 37) with some more details: “The elbow and hip joints were more flexed at impact, at maximal extension and at maximal flexion of the leading limb, whereas the stifle joint was more extended at impact.”

L 410 Please correct reference: McClinchey, H., Thomason, J., and Runciman, R. Grip and slippage of the horse's hoof on solid substrates measured ex Vivo. Biosystems Engineering, 2004, 89(4), S. 485-494. doi:10.1016/j.biosystemseng.2004.08.004

Reviewer #3: PLOS ONE TB SURFACE

17 I’d try to combine the first two sentences as the first one is so general so it doesn’t add to the abstract. Consider

Horseshoes used during racing are a major determinant of safety as they play a critical role in providing traction with the ground surface.

I’d also follow up with a sentence explaining if longer or shorter slip durations were considered optimal or highlight that both too much or too little can cause injury.

31 tracks

34 I’d avoid sensitivity as this is linked to epi and statistics. Is there a better term?

Also, This last sentence is not in past tense and its not clear what the meaning is, please rephrase. How would these findings affect farrier decisions perhaps?

38-40 where are the forelegs? Break downs are commonly foreleg issues

42 please include if a longer or shorter duration of slip is more likely to contribute to injury. or highlight that both too much or too little can cause injury. Maybe move eg of too much and too little refs up closer to the top.

89 so were trials with the same horse but a different jockey counted as different experiences? How did you account for this statistically as I don’t know this would could or should count as different?

45-46 this seems like half of a sentence? Missing a reduced bending moments when?

47 who has shown hoof slip to affect jarring? How would it do that?

48-49 needs to be higher up. Do the refs 1 and 2 support this? Has slip been shown to decrease the rate of energy dissipation?

59 the limited to 3/8 height refers to the shoe or the studs please clarify

69 what is solear? Please clarify?

Intro- generally there wasn’t explicit discussion on the effect of speed and slip which the abstract made it sound as if this was looked at, possibly more should be added here on that. I also think there should be mention of common race horse injuries (break down and otherwise, ideally with commentary on surface type and limb affected) to see where shoeing impacts would be most important.

88i don’t think you need the e.g. and can just leave the ref same for line 101

89 if the pairs were fixed I’m unclear about 14 horse jockey pairings? Does that include shoes? May need a supplemental diagram

103 I know this was previously published, but there should be a brief comment about where on the track the filming occurred aka on a straight segment and how far after the gate. Also a comment on calibration when it seems horses had varied distances from the camera.

125which marker (of the 3seen) on the hoof wand was used? Can you grab an image from the markers at first contact and make it a 4 panel?

Table 1, I am unclear about why the jockey horse pairs differed so much some were 7 and others 12 please explain more in above methods +/-supplemental figure.

I don’t think you need the shoe-surface combo column since you outlined that in the first two columns, this would allow it to be a bit bigger

Table 2 what is the N:? it looks as if speed is significant for the leading forelimb (how is it sig? directionality), with shoe only significant in the leading hindlimb (what shoe type particularly)?? The abstract doesn’t read as such.

Table 3 this needs more clarity in the table legend, is this on slip? This applies to the following tables as well.

Fig 2, discussion should include which of these areas we are trying to optimize slip type in, aka is it more important to have less slip in the leading hind leg because of commonly reported injuries? Or in one of the forelegs? Aka what is the most clinically relevant limb to prioritize an optimal slip type? Or do we just not know?

Add in N

Fig 3, are the outliers all the same horse? Is this across all surfaces? Is it still significant if you just look at the surface individually?

185, but surface was significant, with slip being longer/shorter under what conditions?

Gen comment, I don’t think you have to have the p value to 3 digits beyond the decimal but please adhere to journal rules.

Sentences 190-192 are what I am looking for up in 185

194 if you leave the Pearson correlation in please add to methods. Why does this R2 differ from the one in Fig 4? Is that just for the leading hind limb? If you report this one you should do for the other limbs as well please

Fig 4 here you have P <0.001 but I think this is referring to the line 204 which has p=0.001

It looks like lower speed has more slip- hoping this is discussed.

DISCUSSION

230-231 I’d like this idea to be brought into the conclusions of the abstract vs what is there. I’d still like more discussion of commonly reported injury areas in relation to racing and track types. Aka how often is the leading hind limb injured vs forelimbs etc. are there more hind end soft tissue injuries which would make sense with the greater slip and would agree with prev lit. you touch on it in 264 but for clinical relevance the more commonly injured legs would seem to be the ones you’d pay most attention to in regards to optimizing slip.

275-277 with a longer slip duration for the aluminum shoe vs barefoot this would seem to be the same thing that the jocks were saying where they felt the slip was increased for the aluminum shoe. If you meant these to be in agreement I’d remove the actually as it makes it sound as if the two statements wouldn’t agree. I don’t see the discrepancy? Or is it that it only occurred in one instance vs more? Please clarify.

291-2- were these assessed with the different shoes? Is that a paper to come out soon? Was stride length and frequency calculated in the different shoes?

308-309 also better to be added to the abstract conclusions than what is there. Aka being explicit that slip is hoof specific

Where are your limitations? There was one above re not testing surface moisture and other characteristics but was that really it?

Ref 26 has some odd A figures in it?

I’d love to see the Fig S1actually be in the paper

6. PLOS authors have the option to publish the peer review history of their article (what does this mean?). If published, this will include your full peer review and any attached files.

Reviewer #1: No

Reviewer #2: No

Reviewer #3: No

---

## [Author Response · Author response to Decision Letter 0]

22 Jul 2024

Response to Reviewers

Reviewer #1

Overall this paper provides some data on the slip in different limbs during gallop with 4 different shoeing conditions on 2 different surfaces. These observations are of potential use to underpin investigations into the development of injury in sport horses.

We would like to thank the reviewer for their helpful review. Please find below our point-by-point response to each of their suggestions.

The measurement of hoof slip through kinematics is prone to many errors that were not mentioned or discussed although some of the causes of loss of data were mentioned in lines 120-123.

We have added the following caveat statement to our discussion:

“Using a purely kinematic technique to detect hoof contact and slip-stop may introduce errors, particularly at low frame rates [1,25,47], and the above data indicate quite high variability on measurements.”

As one horse was ridden by 2 jockeys, the number of jockey/horse pairs was given as 14 for the 13 horses. It is not clear to this reviewer how this was treated in the statistics, nor whether that one horse provided data for both jockeys in all the different treatments.

It seems that not all the 13 horses that were used provided data for all the different conditions. Table 1 shows that the number of observations of hoof slip varied between 7 and 19 with more than half (17) below the number of horses for any of the 32 combinations of 4 limbs, 2 surfaces and 4 shoeing types. 

Hence it is clear that not all the 13 horses provided data in all the 32 possible combinations for slip in each of the limbs.

Further, in Table 1 the number of horse jockey pairs that provided data for each of the 8 experimental conditions is below 13 in all but one condition (barefoot artificial).

We explain that “horse-rider pair” was entered as a random factor into the model in the “Statistics” section. There were 14 horse rider pairs. Horse 1, which was used for two jockeys, did not complete all conditions with each jockey. 

For clarity, we now reference one of our other manuscripts (Horan et al., 2021), which details all combinations completed by each horse and rider in Table 1 of that manuscript.

“The shoe-surface combinations completed by each horse-jockey pair are summarised in [29], but please note that video footage was not available for one horse and therefore 14 (rather than 15 horse-jockey pairs, as per [29]) are included in the current study.”

The authors are to be congratulated for clearly providing this data, but it does require some further discussion as to whether the surface characteristics (limitation mentioned in line 254) that were described as varying from soft to good/firm (line99) for the grass showed any relationship with the likelihood of a measurement being acquired. 

We have now included the following information from our previous related manuscripts: 

“Trials took place across multiple days for each horse-jockey pair to acquire data for as many of the eight possible shoe–surface combinations as was feasible; limitations were imposed due to horse and jockey availability and routine turf accessibility restrictions implemented by the BRS to avoid ‘hard’ going [31].” 

We have also clarified: 

“The shoe-surface combinations completed by each horse-jockey pair are summarised in [29], but please note that video footage was not available for one horse and therefore 14 (rather than 15 horse-jockey pairs, as per [29]) are included in the current study.”

This might then better support the statement in Line 308 “However, overall it seems that the specific hoof in question and the surface involved have a greater influence on total slip duration than overall speed.”

Reviewer #2: 

Congratulations on the series of articles. My minor revision decision implies the possible correction of the following aspects:

We would like to thank the reviewer for their comments and suggestions to improve our manuscript.

L 83 I suggest using the International system and converting horse hand to meters.

This has been amended.

L 108-111 and L137 there is a discrepancy between the text (A. Photograph of marker wand on the hoof.) and the additional presence in the photograph of an accelerometer mounted to the hoof.

In the caption, we now clarify that the accelerometer was used in a different study component (Horan et al., 2022).

L 223 “hinlimb” is misspelled

This has been corrected.

L 222-224 For a better accuracy and understanding, I suggest completing the cited information from Back W. et al (reference 37) with some more details: “The elbow and hip joints were more flexed at impact, at maximal extension and at maximal flexion of the leading limb, whereas the stifle joint was more extended at impact.”

Thank you for this suggestion and we looked at how we might add in this extra information. However, as our discussion point was referencing the swing phase rather than the impact we did not feel this was appropriate here. 

L 410 Please correct reference: McClinchey, H., Thomason, J., and Runciman, R. Grip and slippage of the horse's hoof on solid substrates measured ex Vivo. Biosystems Engineering, 2004, 89(4), S. 485-494. doi:10.1016/j.biosystemseng.2004.08.004

This has been corrected.

Reviewer #3: PLOS ONE TB SURFACE

We would like to thank the reviewer for their detailed and helpful suggestions to improve our manuscript.

17 I’d try to combine the first two sentences as the first one is so general so it doesn’t add to the abstract. Consider

Horseshoes used during racing are a major determinant of safety as they play a critical role in providing traction with the ground surface.

Thank you for this suggestion. We now include this as our first sentence.

I’d also follow up with a sentence explaining if longer or shorter slip durations were considered optimal or highlight that both too much or too little can cause injury.

We have added the following:

“Although excessive hoof slip is detrimental and can predispose to instabilities, falls and injuries, some slip is essential to dissipate energy and lower stresses on the limb tissues during initial loading.”

31 tracks

This word has been added.

34 I’d avoid sensitivity as this is linked to epi and statistics. Is there a better term?

We have replaced “sensitivity” with “response”.

Also, This last sentence is not in past tense and its not clear what the meaning is, please rephrase. How would these findings affect farrier decisions perhaps?

We have replaced with the following: 

“This study emphasises the importance of evaluating individual limb biomechanics when applying external interventions that impact the asymmetric galloping gait of the horse. Hoof slip durations and the impact of shoe-surface effects on slip were limb specific. Further work is needed to relate specific limb injury occurrence to these hoof slip duration data.”

38-40 where are the forelegs? Break downs are commonly foreleg issues

We no longer include highlights, as this is not required by PLOS ONE.

42 please include if a longer or shorter duration of slip is more likely to contribute to injury. or highlight that both too much or too little can cause injury. Maybe move eg of too much and too little refs up closer to the top.

We have added that “both too much or too little can cause injury” to our second sentence. We have also considerably restructured our introduction and added additional references on epidemiological studies pertaining to racehorse injuries.

89 so were trials with the same horse but a different jockey counted as different experiences? How did you account for this statistically as I don’t know this would could or should count as different?

We included horse-jockey pair ID as a random factor, as noted in the statistics section.

45-46 this seems like half of a sentence? Missing a reduced bending moments when?

We have adjusted the sentence to include “hoof slip ensures…” to improve clarity.

47 who has shown hoof slip to affect jarring? How would it do that?

Pardoe et al., 2001 mention the impact of slip on jarring. However, we have simplified this sentence to only mention “high frequency oscillations”.

48-49 needs to be higher up. Do the refs 1 and 2 support this? Has slip been shown to decrease the rate of energy dissipation?

We have moved this sentence. Hoof slip permits increased energy dissipation. 

59 the limited to 3/8 height refers to the shoe or the studs please clarify

The 3/8 height refers to the studs. We have clarified this in the manuscript.

69 what is solear? Please clarify?

We have replaced “solear protrusions” with “protrusions from the sole”.

Intro- generally there wasn’t explicit discussion on the effect of speed and slip which the abstract made it sound as if this was looked at, possibly more should be added here on that. 

We have added the following sentence:

“In addition, as hoof accelerations were previously found to show a speed-dependent response to shoe and surface combinations in this sample population [29], we were also interested to investigate how shoe-surface condition might impact slip duration across different gallop speeds.”

I also think there should be mention of common race horse injuries (break down and otherwise, ideally with commentary on surface type and limb affected) to see where shoeing impacts would be most important.

We expanded on our introduction to include these points.

88i don’t think you need the e.g. and can just leave the ref same for line 101

We have amended to the reviewer’s preference, in each case.

89 if the pairs were fixed I’m unclear about 14 horse jockey pairings? Does that include shoes? May need a supplemental diagram

We have now included the following information: 

“The shoe-surface combinations completed by each horse-jockey pair are summarised in [29], but please note that video footage was not available one horse and therefore 14 (rather than 15 horse-jockey pairs, as per [29]) are included in the current study.”

103 I know this was previously published, but there should be a brief comment about where on the track the filming occurred aka on a straight segment and how far after the gate. Also a comment on calibration when it seems horses had varied distances from the camera.

The distance from the cameras was not considered as we were quantifying slip time not slip distance.

We have added that we filmed on a straight segment to the methods: 

Line 128: “We filmed on a straight section of the track, approximately 200 m from the start point.”

125which marker (of the 3seen) on the hoof wand was used? Can you grab an image from the markers at first contact and make it a 4 panel?

The central marker was tracked unless it was obscured in the video, in which case the upper left marker was used. This detail has been added to the manuscript.

We have added an addition panel to the figure with a screenshot of the hoof at first contact, as suggested.

Table 1, I am unclear about why the jockey horse pairs differed so much some were 7 and others 12 please explain more in above methods +/-supplemental figure.

We have now included the following information from our previous related manuscript: 

“Trials took place across multiple days for each horse-jockey pair to acquire data for as many of the eight possible shoe–surface combinations as was feasible; limitations were imposed due to horse and jockey availability and routine turf accessibility restrictions implemented by the BRS to avoid ‘hard’ going [31].” 

We have also clarified: 

“The shoe-surface combinations completed by each horse-jockey pair are summarised in [29], but please note that video footage was not available one horse and therefore 14 (rather than 15 horse-jockey pairs, as per [29]) are included in the current study.”

I don’t think you need the shoe-surface combo column since you outlined that in the first two columns, this would allow it to be a bit bigger

We have deleted this.

Table 2 what is the N:? it looks as if speed is significant for the leading forelimb (how is it sig? directionality), with shoe only significant in the leading hindlimb (what shoe type particularly)?? The abstract doesn’t read as such.

We apologise if this was not clear but note that we do already expand on these points in the text.

First, in the leading forelimb section of the results, we explained that there was a decreasing slip duration with increasing speed; this is also shown in figure 2.

In the abstract, we explained that “In the leading hindlimb, slip durations were also significantly longer for the aluminium condition compared to barefoot, by 3.7 ms.”

Table 3 this needs more clarity in the table legend, is this on slip? This applies to the following tables as well.

We have added that this in “on hoof slip duration” to Tables 3–5.

Fig 2, discussion should include which of these areas we are trying to optimize slip type in, aka is it more important to have less slip in the leading hind leg because of commonly reported injuries? Or in one of the forelegs? Aka what is the most clinically relevant limb to prioritize an optimal slip type? Or do we just not know?

We have added the following to the discussion:

“In addition, at present, we do not know the most clinically relevant limb in which to prioritise optimal slip type. The exact nuances of the likelihood of injury are a complex interplay of various additional race characteristics including, but not limited to, horse age, sex, race distance and field size [7,18,19]. However, given that forelimbs appear most likely to fracture, emphasis might initially be placed on further investigations into the forelimbs.”

Add in N

The full number of observations per condition are available in Table 1. 

Fig 3, are the outliers all the same horse? Is this across all surfaces? Is it still significant if you just look at the surface individually?

No, the outliers were for different horses:

• The outlier for the leading fore was Horse 11 (Horse-jockey pair 11): GluShu on turf

• The outlier for the non-leading forelimb was Horse 10 (Horse-jockey pair 10): steel on turf

• The outlier for the aluminium condition in the non-leading hind is Horse 13 (Horse-jockey pair 13): barefoot on turf.

• The outlier for the barefoot condition in the non-leading hind is Horse 1 (Horse-jockey pair 1): barefoot on turf.

We have added a note to the captions of Figs 2 and 3, explaining that outliers came from different horses. In addition, we have added the following sentence to our discussion: “All of the outliers in Figs 2 and 3 represent data collected on turf surfaces, suggesting the turf was more prone to extreme variations in hoof slip.”

The mixed model approach evaluates surface (and shoe-type) individually and the surface effects are shown in Figure 2.

185, but surface was significant, with slip being longer/shorter under what conditions?

Yes, and we explain on line 186 (original version) that “The estimated marginal means for surface effects indicated that slip duration was 3.2 ms longer on turf.”

We have now clarified “…longer on turf than on the artificial surface” but we are unclear what additional information the reviewer requires here?

Gen comment, I don’t think you have to have the p value to 3 digits beyond the decimal but please adhere to journal rules.

PLOS ONE does not appear to specify the precision required on p values, so we have left this as is.

Sentences 190-192 are what I am looking for up in 185

Please see response above.

194 if you leave the Pearson correlation in please add to methods. Why does this R2 differ from the one in Fig 4? Is that just for the leading hind limb? If you report this one you should do for the other limbs as well please

Figure 4 is for the leading forelimb. The R2 reported in line 194 is for the leading hindlimb. 

We have now included a four-part figure, showing the relationship between slip and speed for all limbs. Figure 4 is based on the raw data. 

Fig 4 here you have P <0.001 but I think this is referring to the line 204 which has p=0.001

It looks like lower speed has more slip- hopi

---

## [Decision Letter · Decision Letter 1]

19 Aug 2024

PONE-D-24-21823R1Hoof slip duration at impact in galloping Thoroughbred ex-racehorses trialling eight shoe-surface combinationsPLOS ONE

Dear Dr.
Horan,

Thank you for submitting your manuscript to PLOS ONE. After careful consideration, we feel that it has merit but does not fully meet PLOS ONE’s publication criteria as it currently stands. Therefore, we invite you to submit a revised version of the manuscript that addresses the points raised during the review process.

We look forward to receiving your revised manuscript.

Kind regards,

Tzen-Yuh Chiang

Academic Editor

PLOS ONE

Journal Requirements:

Reviewers' comments:

Reviewer's Responses to Questions

**Comments to the Author**

1. If the authors have adequately addressed your comments raised in a previous round of review and you feel that this manuscript is now acceptable for publication, you may indicate that here to bypass the “Comments to the Author” section, enter your conflict of interest statement in the “Confidential to Editor” section, and submit your "Accept" recommendation.

Reviewer #2: All comments have been addressed

Reviewer #3: (No Response)

2. Is the manuscript technically sound, and do the data support the conclusions?

Reviewer #2: Yes

Reviewer #3: Yes

3. Has the statistical analysis been performed appropriately and rigorously? 

Reviewer #2: Yes

Reviewer #3: Yes

4. Have the authors made all data underlying the findings in their manuscript fully available?

Reviewer #2: Yes

Reviewer #3: Yes

5. Is the manuscript presented in an intelligible fashion and written in standard English?

Reviewer #2: Yes

Reviewer #3: Yes

6. Review Comments to the Author

Reviewer #2: (No Response)

Reviewer #3: thanks for the changes.

minor comments

PLOS one_hoof slip_2024_revision1

62 spellout US

65 spell out UK

Fig 3, the star is just over the aluminumshoes in the leading hind, more detail in the figure legend should be given to understand what comparison is significant.

Fig 5 discussion of statistical significance and a correlation vs a true impact should be discussed as the R^2 are exceptionally weak.

7. PLOS authors have the option to publish the peer review history of their article (what does this mean?). If published, this will include your full peer review and any attached files.

Reviewer #2: No

Reviewer #3: No

---

## [Author Response · Author response to Decision Letter 1]

6 Sep 2024

Reviewer #3: 

thanks for the changes.

minor comments

62 spellout US

This has been amended.

65 spell out UK

This has been amended.

Fig 3, the star is just over the aluminumshoes in the leading hind, more detail in the figure legend should be given to understand what comparison is significant.

We have added additional detail to the legend as requested.

Fig 5 discussion of statistical significance and a correlation vs a true impact should be discussed as the R^2 are exceptionally weak.

We have adjusted the discussion paragraph on this figure to further emphasise the weak correlation: “It is also worth noting that our models identified no significant effect of speed on hoof slip duration for the non-leading forelimb and hindlimbs, and there was only a weak negative correlation for the leading forelimb (Fig 5). The observation of a reduction in slip duration in the leading forelimb as speed increases may be related to the fact that at higher speeds there is less time available for the hoof to be in contact with the ground. However, given that the correlation is weak, the true impact of this relationship is likely to be minor. Instead, it seems that the specific hoof in question and the surface involved have a greater influence on total slip duration than overall speed.”

---

## [Decision Letter · Decision Letter 2]

27 Sep 2024

Hoof slip duration at impact in galloping Thoroughbred ex-racehorses trialling eight shoe-surface combinations

PONE-D-24-21823R2

Dear Dr. Horan,

We’re pleased to inform you that your manuscript has been judged scientifically suitable for publication and will be formally accepted for publication once it meets all outstanding technical requirements.

Kind regards,

Tzen-Yuh Chiang

Academic Editor

PLOS ONE

Additional Editor Comments (optional):

Reviewers' comments:

Reviewer's Responses to Questions

**Comments to the Author**

1. If the authors have adequately addressed your comments raised in a previous round of review and you feel that this manuscript is now acceptable for publication, you may indicate that here to bypass the “Comments to the Author” section, enter your conflict of interest statement in the “Confidential to Editor” section, and submit your "Accept" recommendation.

Reviewer #2: All comments have been addressed

Reviewer #3: All comments have been addressed

2. Is the manuscript technically sound, and do the data support the conclusions?

Reviewer #2: Yes

Reviewer #3: Yes

3. Has the statistical analysis been performed appropriately and rigorously? 

Reviewer #2: Yes

Reviewer #3: Yes

4. Have the authors made all data underlying the findings in their manuscript fully available?

Reviewer #2: Yes

Reviewer #3: Yes

5. Is the manuscript presented in an intelligible fashion and written in standard English?

Reviewer #2: Yes

Reviewer #3: Yes

6. Review Comments to the Author

Reviewer #2: (No Response)

Reviewer #3: Thanks for your changes. no concerns left

I"m not sure why this won't let me submit without adding more.

7. PLOS authors have the option to publish the peer review history of their article (what does this mean?). If published, this will include your full peer review and any attached files.

Reviewer #2: No

Reviewer #3: No

---

## [Editor Report · Acceptance letter]

2 Oct 2024

PONE-D-24-21823R2 

PLOS ONE

Dear Dr. Horan, 

I'm pleased to inform you that your manuscript has been deemed suitable for publication in PLOS ONE. Congratulations! Your manuscript is now being handed over to our production team.

Kind regards, 

on behalf of

Dr. Tzen-Yuh Chiang 

Academic Editor

PLOS ONE